# Variations in elemental composition of rice (*Oryza sativa* L.) with different cultivation areas of Ethiopia

**Abebe Desalew**[1,2]**, Bewketu Mehari**[1]*

**1** Department of Chemistry, College of Natural and Computational Sciences, University Of Gondar, Gondar, Ethiopia, **2** Department of Chemistry, College of Natural and Computational Sciences, Mizan Tepi University, Mizan Teferi, Ethiopia

* bewketu.mehari@uog.edu.et

**Data Availability Statement:** All relevant data are within the paper.

**Funding:** The author(s) received no specific funding for this work.

## Abstract

Variations in the elemental composition of rice (*Oryza sativa* L.) grains, and the link with the growing soil, were investigated across the major production areas of Ethiopia (Fogera, Metema and Pawe). The elements (Ca, Mg, Fe, Zn, Mn, Cu, Ni, Cr, Cd and Pb) were determined by using flame atomic absorption spectroscopy (FAAS), after digesting samples through an optimized procedure with respect to volumes of reagents ($HNO_3$, $HClO_4$ and $H_2O_2$), temperature and time. The accuracy of the FAAS method was in the range of 87–113%. The most abundant element in rice was Mg (414–561 mg kg$^{-1}$) followed by Fe (49.4–168 mg kg$^{-1}$), while in soil was Fe (11674–12917 mg kg$^{-1}$) followed by Mg (619–709 mg kg$^{-1}$). Chromium, Cd and Pb were all below the limit of quantitation of the method. The concentrations of the elements, except Zn in rice and Fe in soil, varied significantly ($p < 0.05$) with the growing region. Notably, rice from Fogera contained more than double Fe, while from Pawe less than half Cu than from the other region. Soils from the rice fields of Pawe, generally, had lower levels of the elements than from the other regions. The order of the abundances of the elements in soil was reflected in the rice grains, except for the reversal between Fe and Mg. However, elemental concentrations were higher in soil than in rice, indicating the absence of bioaccumulation by the rice grains. Furthermore, only copper exhibited a strong positive correlation ($r = 0.991$) between the rice grain and soil.

## Introduction

Rice is a plant that belongs to the family Poaceae (Gramineae) and the genus *Oryza*. There are about 25 species of the genus *Oryza*. Of these only two species, *O. sativa* and *O. glaberrima*, are under commercial cultivation. The species *O. sativa* is believed to have originated from Southeast Asia and spread to different parts of the world, while *O. glaberrima* is still confined to its original place in West Africa [1].

Rice is a staple food crop for most of the world's population [2], and it stands as the third most prevalent cereal crop next to wheat and maize [3,4]. Rice is also the most rapidly growing

**Competing interests:** The authors have declared that no competing interests exist.

source of food, where it presents its importance to food security and food self-sufficiency for poor countries including in Africa [2,5].

In contrast, rice is not a staple food crop in Ethiopia, where its production started only a few decades ago [6]. Recently, however, a considerable proportion of the country's population has started to recognize its use as a food crop [1]. Furthermore, the crop is becoming more valued for a variety of uses, in the preparation of various local foods, such as bread, porridge, couscous and soup, and alcoholic beverages, such as beer and liquor, either alone or mixed with other crops [5]. Since 2006, Ethiopian rice production trends show an increase in both area and productivity and currently, Fogera, Metema and Pawe are the major rice-producing areas in Ethiopia [7].

Metallic elements are present in the environment in different proportions, where some of them play important roles in plants and animals and are considered essential, while others are deemed non-essential or potentially toxic even at trace levels [2]. The essential elements include Mg, Ca, Fe, Zn, Mn, Cu and Zn, while the non-essential and potentially toxic elements include Cd and Pb. Nutritionally, based on the amounts needed by plants and animals, the essential elements are further classified as major, such as Mg and Ca, and trace, such as Fe, Zn, Mn, Cu and Zn, essential elements [8]. Analysis of the elemental composition of plant material is, therefore, important from the essential, nutritional and toxicological perspectives [9]. In addition to the plant materials, the analysis of the growing soil is important to identify nutrient deficiency or excess as well as the link between the composition of the plant and the soil [10]. The elemental composition of a plant material mainly varies with the different botanical varieties of the plant and its growing environmental conditions, including the elemental composition of the growing soil [11]. Studies conducted on the chemical compositions of rice cultivated in upland [12] and lowland [13] areas in Ethiopia, have indicated the presence of significant differences in rice quality attributes. Meharg et al. [14] have observed global variations in the elemental compositions of rice cultivated in different continents and countries. Uptake by plants from soil is a major source of metals in plant-based foods for human consumption [15,16]. This study aimed to determine the concentrations of essential and non-essential metals in rice (*Oryza sativa* L.) grains and assess their variations among the major production regions of Ethiopia. The correlation between the elemental concentrations in the rice grain with that in the growing soil was also investigated.

## Materials and methods

### Description of the study area

The study was conducted on the three major rice production regions of Ethiopia (Fogera, Metema and Pawe) (Fig 1). Fogera is located in the South Gondar Zone of Amhara Regional State; Metema is located in the West Gondar Zone of Amhara Regional State, and Pawe is located in the Metekel Zone of Benshangul Gumz Regional State.

### Rice samples

From each region, three sampling areas were selected based on the extent of rice cultivation. The areas were Kuahr, Work Meda and Avua Kokit (Fogera); Shinfa, Das and Kokit (Metema); and Almu, Felege Selam and Hidase (Pawi). From each area, three samples, each of 0.5 kg rice grains, were collected from three different farmers and mixed together. Samples were collected from private farmlands, after permission was granted by the farmers to access the fields. The samples were packed into clean polyethylene plastic bags, labeled and stored at room temperature in the laboratory until analysis.

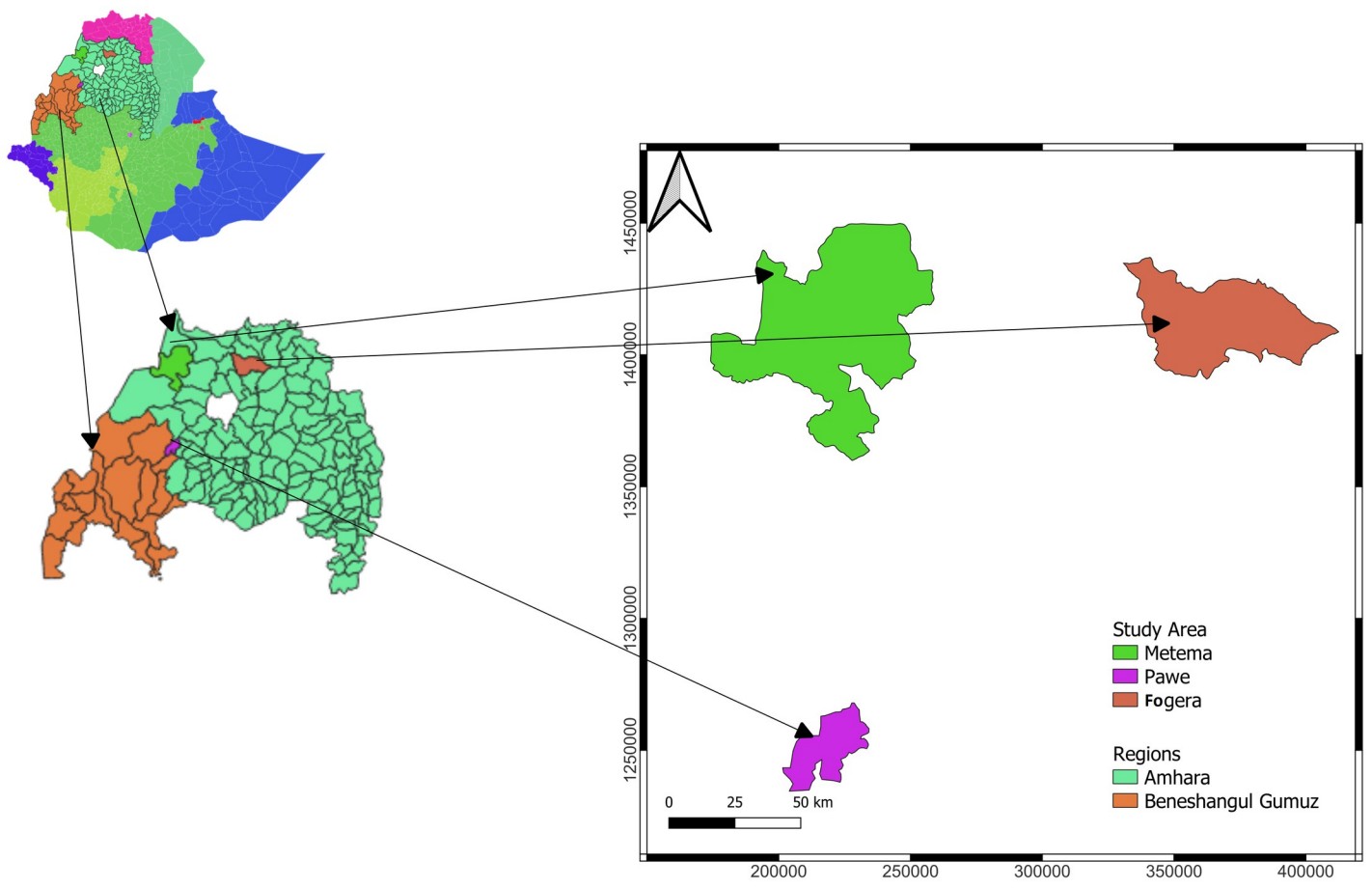

**Fig 1. Map showing the major rice producing areas of Ethiopia.**

## Soil samples

Soil composite samples were collected from the farmlands where the rice samples were grown. The soil samples were collected from the surface, within 15–20 cm depth. Corresponding to each sampling region, three samples, each of 1.5 kg soil, were collected from different farmlands. The samples were packed in polyethylene plastic bags, labeled, and stored at room temperature in the laboratory until analysis.

## Equipment

Flame atomic absorption spectrometer (Buck Scientific, 210VGP, USA) with air-acetylene flame, hot plate (SH3, Steriline Ltd, UK), electronic balance (Citizone, CTG 1200–1200, India), blender (Ika-Werke, Germany) were used in the study. A refrigerator (Hitach LR902T, England) was also used to keep digested samples until analysis.

## Chemicals

Nitric acid (69%), perchloric acid (70%), lanthanum chloride hydrate, standard solutions (1000 mg/L) of Ca, Mg, Fe, Zn, Cr, Cu (Blulux laboratories, India), $H_2O_2$ (30%) (Okhla,

India), standard solutions (1000 mg/L) of Ni, Mn, Cd and Pb (Loba Chemie, India) were used in the study.

## Sample preparation

### Sample pretreatment

The rice samples were washed with tap water followed by distilled water to remove dust materials from the surface of the grains, and dried until constant weights. The dried rice samples were then powdered and sieved to pass through a 0.5 mm mesh. The soil samples were first air-dried to constant weights for three days and sieved through a 2 mm mesh to remove large debris and stones. The samples were then powdered, sieved through a 0.5 mm mesh and homogenized.

### Sample digestion

The powdered rice and soil samples were digested following the acid digestion method, after optimization to the volumes of $HNO_3$ (69%), $HClO_4$ (70%) and $H_2O_2$ (30%) in mixtures, digestion time and digestion temperature. Applying the optimized conditions, 1 g each of powdered rice and soil was transferred into two flasks. Then 7 and 9 mL of $HNO_3:HClO_4:H_2O_2$ mixtures were added in the ratio by volume of 4:2:1 to rice and 4:3:2 to the soil, respectively. The samples were digested for 1 h at 160°C for rice and for 2:30 h at 240°C for soil placed on a hot plate. The digest was cooled and 5 mL of distilled water was added. Then, the solution was filtered with Whatman no.1 filter paper into a 50 mL volumetric flask and 1 mL of 1% lanthanum chloride hydrate ($LaCl_3.7H_2O$) was added and the volume made up to the mark with distilled water. Reagent blank solutions were also prepared following the procedures used for the digestion of the rice and soil samples.

### FAAS determination of elements

The concentrations of elements in the digested rice and soil samples were determined by using a flame atomic absorption spectrophotometer (FAAS) equipped with a deuterium arc background corrector and air-acetylene flame.

The atomic absorption spectrometer was calibrated using six-point standard solutions, corresponding to each element, in the concentration (mg $L^{-1}$) range of 2.0–12.0 for Mg, 0.40–12 for Ca, 0.2–17 for Fe, 0.1–7.0 for Cu, 0.2–4.0 for Zn, 0.1–12.0 for Mn, 0.1–4.0 for Ni, 0.027–8.0 for Cd, 0.02–5.0 for Cr and 0.2–8.0 for Pb.

### Method validation

Accuracy and limits of detection were determined to assess the validity of the methods used for the digestion and analysis of the rice and soil samples. The accuracy of the method was determined by spiking the samples with known concentrations of standard solutions and submitting them to the preparation and analysis procedure used for the samples. The accuracy was calculated as percentage recovery values. The limit of detection (LOD) of the method was determined from the measurement of three blank samples, which were digested and analyzed along with each sample. The LOD of the method was calculated as three times the standard deviation of the blank signal divided by the slope of the calibration equation. Similarly, the limit of quantitation (LOQ) of the method was calculated as ten times the standard deviation of the blank signal divided by the slope of the calibration equation.

**Table 1. Optimization of reagent volumes for the digestion of rice (1 g, 200˚C, 2 h) in the presence of 1 mL $H_2O_2$ and soil (1 g, 240˚C, 2:30 h) in the presence of 2 mL $H_2O_2$.**

| Rice | | | Soil | | |
|---|---|---|---|---|---|
| Volume (mL) | | Appearance of solution | Volume (mL) | | Appearance of solution |
| $HNO_3$ | $HClO_4$ | | $HNO_3$ | $HClO_4$ | |
| 1 | 1 | Brown turbid | 1 | 0 | White with residue |
| 0 | 2 | Yellow dried matter | 0 | 1 | Brown dried matter |
| 2 | 0 | Yellow dried matter | 2 | 0 | White with residue |
| 1 | 2 | Yellow with some residue | 0 | 2 | White dried matter |
| 2 | 1 | Milky White with residue | 3 | 0 | White with residue |
| 3 | 0 | Yellow turbid | 0 | 3 | Yellow turbid |
| 0 | 3 | Yellow dried matter | 1 | 3 | Clear with residue |
| 4 | 0 | Yellow turbid | 2 | 3 | Milky white |
| 0 | 4 | Yellow turbid | 3 | 3 | Milky white |
| 5 | 0 | Yellow turbid | 4 | 3 | *Clear and colorless* |
| 0 | 5 | Light yellow | | | |
| 4 | 1 | Milky white with residue | | | |
| 4 | 2 | *Clear and colorless* | | | |

## Determination of soil pH

A 5 g of each ground soil sample was added into a 50 mL beaker and mixed with 10 mL of distilled water. The solution was stirred for 30 s and waited for 3 min, for a total of five stirring and waiting cycles. Then the mixture was allowed to settle for about 5 min until a supernatant, clear liquid above the settled soil, formed. Finally, by inserting the probe of the pH meter into the supernatant, the pH of the soil sample was measured.

# Results and discussion

## Optimum conditions for sample digestion

Powdered rice and soil samples required different digestion conditions to produce clear and colorless solutions suitable for FAAS analysis (Tables 1–3).

## Analytical characteristics of the method

The concentrations of the elements were determined by using calibration curves constructed from a series of standard solutions of each element and the corresponding FAAS absorbance readings. The correlation coefficients of the calibration curves were greater than 0.998 for all the analyzed elements (Table 4), indicating good linearity of the method. The limits of detection (LOD) of the method were in the range of 1.0–22.9 mg kg$^{-1}$ across the different elements

**Table 2. Optimization of temperature for the digestion of rice (1 g, 2 h) and soil (1 g, 2:30 h).** Reagents: 4 mL $HNO_3$, 2 mL $HClO_4$, 1 mL $H_2O_2$ for rice; 4 mL $HNO_3$, 3 mL $HClO_4$, 2 mL $H_2O_2$ for soil.

| Rice | | Soil | |
|---|---|---|---|
| T (˚C) | Appearance of solution | T (˚C) | Appearance of solution |
| 80 | Yellowish turbid | 120 | Gray |
| 120 | Clear light yellow | 160 | Light gray |
| 160 | *Clear and colorless* | 200 | Clear with residue |
| | | 240 | *Clear and colorless* |

**Table 3. Optimization of time for the digestion of rice (1 g, 160°C) and soil (1 g, 240°C).** Reagents: 4 mL $HNO_3$, 2 mL $HClO_4$, 1 mL $H_2O_2$ for rice; 4 mL $HNO_3$, 3 mL $HClO_4$, 2 mL $H_2O_2$ for soil.

| Rice | | Soil | |
|---|---|---|---|
| Time (min) | Appearance of solution | Time (min) | Appearance of solution |
| 20 | Yellowish turbid | 90 | Light white with much turbid |
| 30 | Clear light yellow | 105 | Light white with little turbid |
| 40 | Clear with white fume | 120 | Colorless with white fume |
| 50 | Clear with white fume | 150 | **Clear and colorless** |
| 60 | ***Clear and colorless*** | | |

and samples (Table 4). Similarly, the limits of quantitation (LOQ) of the method were in the range of 3.3–89 mg kg$^{-1}$ across the different elements and samples. The LOD and LOQ of the method are low enough to be useful for the detection and quantification of the elements even at trace levels in the samples. The percentage recoveries were all within the range 86.8±8.9–111 ±10% across the different elements and sample types, confirming the validity of the digestion procedure used for the analysis of the elements in the samples.

## Concentrations of metals in rice

The elements Ca, Mg, Cr, Mn, Fe, Ni, Cu, Zn, Cd and Pb were analyzed in the rice samples. Among these, Ca, Mg, Mn, Fe, Cu and Zn were determined quantitatively in all of the rice samples (Table 5). Even though they were detected, the elements Cr, Ni and Cd were all below the LOQ of the method and hence were not quantified accurately. In contrast, Pb was not detected in any of the rice samples.

Magnesium was the most abundant element found in the rice grains, followed by Fe and Ca. The higher concentration of Mg found in rice seeds can be explained by the fact that it is a major component of plant nutrients [6], where it is relatively more abundant in the parts of plants concerned with vital processes, such as seeds and foliage, than in storage parts such as stems and roots [8]. The determined concentration of Mg in rice from Pawe (average 546 mg kg$^{-1}$) and Fogera (561 mg kg$^{-1}$) are within the range 504–1209 reported for rice varieties from Ethiopia [12,13].

Regarding the distributions of the two major essential metals (Ca and Mg) among the three regional samples, rice grains from Fogera and Pawe were found to contain comparable

**Table 4. The wavelength, coefficient of determination (r$^2$), the limit of detection (LOD), the limit of quantitation (LOQ) and recovery values for the method used for the determination of elements in rice and soil using flame atomic absorption spectroscopy.**

| Element | Wavelength (nm) | r$^2$ | LOD (mg kg$^{-1}$) | | LOQ (mg kg$^{-1}$) | | %Recovery | |
|---|---|---|---|---|---|---|---|---|
| | | | Rice | Soil | Rice | Soil | Rice | Soil |
| Mg | 285.2 | 0.9983 | 11.0 | 18.5 | 36.7 | 61.7 | 98.1±3.7 | 109.6±0.9 |
| Ca | 422.7 | 0.9999 | 13.5 | 1.0 | 45.0 | 3.3 | 91.7±7.2 | 92.2±8.0 |
| Cr | 357.9 | 0.9990 | 12.5 | 5.0 | 41.7 | 16.7 | 88.2±0.1 | 111±10 |
| Mn | 279.5 | 0.9995 | 4.0 | 4.0 | 13.3 | 13.3 | 100±7.9 | 91.4±1.8 |
| Fe | 248.3 | 0.9988 | 6.9 | 22.9 | 26.7 | 89.1 | 100±7.9 | 87.2±0.1 |
| Ni | 232.0 | 0.9981 | 3.0 | 1.5 | 10.0 | 5.0 | 97.6±13 | 95.3±3.5 |
| Cu | 342.7 | 0.9984 | 2.0 | 3.0 | 6.7 | 10.0 | 90.5±9.3 | 89.8±0.4 |
| Zn | 213.9 | 0.9983 | 2.5 | 1.0 | 8.3 | 3.3 | 95.6±2.7 | 112.8±0.1 |
| Cd | 228.9 | 0.9996 | 1.0 | 1.0 | 3.3 | 3.3 | 86.8±8.9 | 91.58±8.3 |
| Pb | 283.2 | 0.9987 | 4.5 | 5.5 | 15.0 | 18.3 | 98.1±3.7 | 109.6±0.9 |

**Table 5. The concentration (mg kg$^{-1}$ dry weight) of the elements was determined in the rice and soil samples from the three major cultivation areas of Ethiopia.**

| Element | Rice | | | Soil | | |
|---|---|---|---|---|---|---|
| | Fogera | Metema | Pawe | Fogera | Metema | Pawe |
| Ca | 45.2±3.5 | 57.7±4.5 | 48.3±3.8 | 440±38 | 655±57 | 282±24 |
| Mg | 561±21 | 414±16 | 546±21 | 709±15 | 708±24 | 619±23 |
| Cr | <LOQ | <LOQ | <LOQ | <LOQ | <LOQ | <LOQ |
| Mn | 40.2±3.2 | 27.4±2.2 | 29.8±2.4 | 295±5.8 | 417±31 | 169±25 |
| Fe | 168±13 | 71.8±5.7 | 49.4±3.9 | 12618±1036 | 12917±946 | 11674±515 |
| Ni | <LOQ | <LOQ | <LOQ | 9.69±0.36 | 11.9±0.44 | <LOQ |
| Cu | 33.8±3.5 | 41.7±4.3 | 12.0±1.2 | 85.5±13 | 140±15 | 60.0±13 |
| Zn | 28.1±3.2 | 24.2±2.9 | 24.4±3.1 | 55.8±4.6 | 52.3±2.6 | 26.6±4.7 |
| Cd | <LOQ | <LOQ | <LOQ | <LOQ | <LOQ | <LOQ |
| Pb | <LOD | <LOD | <LOD | <LOQ | <LOD | <LOQ |

*<LOD is below the limit of detection, and <LOQ is below the limit of quantitation of the method.

concentrations with each other. On the other hand, rice from Metema contained significantly higher Ca and lower Mg than that from the other regions.

The average concentration of Ca quantified in this study (45.2–57.7 mg kg$^{-1}$) falls within the range of 10.5–129.4 mg kg$^{-1}$ reported for Brazilian rice grains by Batista et al. [17], and 6–159 mg kg$^{-1}$ reported for lowland rice in Ethiopia by Abera et al. [13]. On the other hand, the determined Ca level is much higher than that (6.2 ±2.7 mg kg$^{-1}$) reported for rice from Iran by Falahi et al. [18]. In contrast, it is lower than the range 77–273 mg kg$^{-1}$ reported by Abera et al. [14].

Iron was found to be the highest in concentration among the essential trace elements (Cr, Mn, Fe, Ni and Cu) analyzed in the rice samples. There were statistically significant differences among the regional rice samples in their Fe contents. Rice grains from Fogera contained higher concentrations of Fe (average 168 mg kg$^{-1}$) than that from Metema (average 71.8 mg kg$^{-1}$) and Pawe (average 49.4 mg kg$^{-1}$). The observed variations were huge, with the amount measured in Fogera rice exceeds from the others by a factor of two or more. Excess Fe in the body causes various health problems including hemochromatosis, liver cirrhosis and membrane lipid damage [19,20]. The Fe contents of rice grains from Pawe (average 49.4 mg kg$^{-1}$) and Metema (average 71.9 mg kg$^{-1}$) are within the range of 9.0–92.6 mg kg$^{-1}$ of Fe reported for rice from Ethiopia [12,13]. The levels of Fe determined in all of the rice samples, however, were below the FAO/WHO [21] recommended maximum permissible limit for Fe (425 mg kg$^{-1}$) (Table 6).

The measured concentration of manganese was in the range of 27.4–40.2 mg kg$^{-1}$ across the rice samples cultivated in the three regions. The measured Mn concentrations are within the range 8.9–40.2 mg kg$^{-1}$ reported for rice from Ethiopia [12,13]. Rice grown in Fogera contained significantly higher concentrations of Mn than that grown in Metema and Pawe.

**Table 6. Standards for the maximum safe limit (mg kg$^{-1}$) of elements in rice and soil.**

| Element | Ca | Mg | Cr | Mn | Fe | Ni | Cu | Zn | Cd | Pb |
|---|---|---|---|---|---|---|---|---|---|---|
| **Rice** | 1000 | – | 2.3 | 500 | 425 | 67 | 73 | 99 | 0.2 | 0.3 |
| **Soil** | – | – | 100[a] | 2000[a,b] | 50000[a,b] | 50[c] | 100[c] | 300[c] | 3[c] | 100[c] |

[a]Source: FAO/WHO [21]

[b]Source: Kabata-Pendias [22]

[c]Source: Ewers [23]. The source of data corresponding to rice is FAO/WHO [21].

Manganese is an essential element required for various biochemical processes. Mn is essential for the normal bone structure, reproduction and normal functioning of the central nervous system. Its deficiency also causes reproductive failure in both males and females [24]. On the other hand, excess intake of Mn causes reproductive deficits, skeletal abnormalities, lethargy and mental disturbances [25].

The concentrations of copper in the rice samples were in the range of 12.0–41.7 mg kg$^{-1}$ across the different regional rice samples. Rice from Pawe contained significantly lower amounts (average 12.0 mg kg$^{-1}$) of Cu than from Metema (average 41.7 mg kg$^{-1}$) and Fogera (average 33.8 mg kg$^{-1}$). Copper is essential for our body, including as a co-factor in enzymes, a catalyst for heme synthesis and iron absorption. However, an excess amount of Cu in the body causes hemolytic anemia and hepatic cirrhosis [26]. The amount of Cu found in all of the regional rice samples was lower than the maximum permissible limit (73 mg kg$^{-1}$) in cereals recommended by FAO/WHO [21].

The concentration of Cu found in the rice samples (12.0–41.7 mg kg$^{-1}$) is higher than that reported for rice from Nigeria [4], China [27] and Korea [28]. In contrast, the amount of Cu described for rice from India (52.72 mg kg$^{-1}$) by Sharma and Raju [29] is higher than the amount determined in this study. On the other hand, the level of Cu reported for rice from Iran (22.8 mg kg$^{-1}$) by Falahi et al. [18] falls within the range of this study.

Unlike the other determined trace elements, there was no statistically significant difference in the concentration of Zn among the three regional rice grains. The measured concentration of zinc ranged from 24.2–28.1 mg kg$^{-1}$ across the regional rice samples. Zinc was the least abundant of the determined trace elements in the rice grains, except for Pawe rice in which the minimum concentration was measured for Cu. Zinc is an essential element to humans and has an important role in metabolism. It is a co-factor for various enzymes in the body. Zinc deficiency leads to coronary heart diseases and metabolic disorders [24]. Despite its importance, excess Zn in the body can cause diarrhea, vomiting and neurological damage [30,31]. The permissible level of Zn set by FAO/WHO [21] is 99.4 mg kg$^{-1}$. The amount of Zn found in the rice samples was below the permissible limit set by FAO/WHO [21].

The determined concentration of Zn (24.2–28.1 mg kg$^{-1}$) was within the range 16.4–140 mg kg$^{-1}$ reported for commercially available rice in Ethiopia by Tegegne et al. [6], and 13.2–26.0 mg kg$^{-1}$ upland [12] and 17.7–35.6 mg kg$^{-1}$ lowland [13] rices in Ethipia. Additionally, the results agree with the amount of Zn indicated for rice from Iran (28.6 mg kg$^{-1}$) by Falahi et al. [18]. On the other hand, the determined concentration is higher than that (0.05–14.87 mg kg$^{-1}$) reported by Emumejaye [4] for rice grains from Nigeria.

In summary, except forZn, the concentrations of all the determined elements varied significantly among the three regional rice samples. Notably, Fogera rice contained more than twice as Fe as the others, while Pawi rice contained less than half as Cu as the others. As the agronomic practices are similar among the growing regions, where rice is predominantly produced by small-scale farmers using traditional farming systems with products normally considered organic, the observed variations may be attributed to differences like the growing soil and climate conditions.

## Concentrations of metals in soil

All of the analyzed metals (Ca, Mg, Cr, Mn, Fe, Ni, Cu, Zn, Cd and Pb) were detected in the soil samples, except Pb, which was not detected in soil samples from Metema. However, in all of the samples, Cr, Cd and Pb were below the LOQ of the method and were not quantified. Additionally, Ni was not found in quantifiable amounts in soil samples from Pawe.

The results revealed that the analyzed metals are present in the soils at different concentrations. The concentration of iron was the highest among the metals analyzed in soil samples

from the three rice growing areas studied (Table 5). Across the sample areas, the level of Fe was found to be in the range of 11674–12917 mg kg$^{-1}$. The soil samples exhibited high and comparable concentrations without statistically significant differences across the rice-growing regions. The high level of iron might be due to its abundance, as it is the fourth most abundant element in the earth's crust that is normally present in all soils [32].

The amount of Fe determined in the soil samples falls within the range 9947–27427 mg kg$^{-1}$ reported for soil from the rift valley region of eastern Shoa, Ethiopia, by Bedassa [24]. On the other hand, the determined concentrations of Fe are lower than that (23866–32262 mg kg$^{-1}$) reported for soils used for the cultivation of garlic in Gojjam, Ethiopia, by Addis and Abebaw [32]. Contrary to these, the measured amounts of Fe are significantly higher than soil samples from Turkey (7368±135 mg kg$^{-1}$) reported by Tuzen [33] and soil samples from Nigeria (1771 ±112 mg kg$^{-1}$) reported by Yusuf et al. [34].

The second most abundant element next to Fe was Mg. The average concentration of Mg determined in the studied soil samples was in the range of 619–709 mg kg$^{-1}$. The third abundant element found in the soil samples was Ca. Significant differences were observed in the Ca contents of soils from the three regions. Soil samples from Metema contained significantly higher concentrations of Ca (average 655 mg kg$^{-1}$) than that from Fogera (average 440 mg kg$^{-1}$) and Pawe (average 282 mg kg$^{-1}$). The higher concentration of Ca found in the soil samples from Metema agrees with the higher concentration of Ca found in rice samples from the region. This observation confirms, among others, the fact that the distribution of metals in plant seeds is a reflection of the mineral composition of the soil in which the plant grows [35].

The levels of Mg and Ca found in the studied soil samples are higher than the amounts 0.82–2.477 mg kg$^{-1}$ Mg and 0.51–1.02 mg kg$^{-1}$ Ca reported by Fenta and Kidanemariam [10] for soil used for the cultivation of khat in Southern Region, Ethiopia.

The next abundant metal to Ca was Mn, with concentrations varied in the range of 169–417 mg kg$^{-1}$ across the regional soil samples. The regional soil samples also exhibited significant differences among each other in their Mn contents. Once again, soil samples from Metema contained significantly higher concentrations of Mn (average 417 mg kg$^{-1}$) than that from Fogera (average 295 mg kg$^{-1}$) and Pawe (average 169 mg kg$^{-1}$). Unlike the observation with Ca, the higher amount of Mn found in Metema soil was not accurately reflected by its relative amount in the rice samples, where it was exceeded by rice samples from Fogera. Soils with a pH less than 5.5 have been indicated to contain a large proportion of manganese as Mn$^{2+}$ in the water-soluble and exchangeable form. With increasing soil pH, Mn$^{2+}$ is converted into its higher oxides (Mn$^{3+}$ and Mn$^{4+}$) which are insoluble in water and therefore, unavailable to plants [10]. Hence, the low pH values measured (4.30–4.95) together with the considerable amount of Mn found in the soil samples indicate the high availability of Mn for plant uptake in the studied rice farming areas.

The level of Mn found in the soil samples is generally lower than the level of Mn found in soils used for the cultivation of garlic in Gojjam, Ethiopia, (402–584 mg kg$^{-1}$) [32], as well as in soil from rift valley region of eastern Shoa, Ethiopia, (789–1546 mg kg$^{-1}$) [24].

The determined average concentration of copper in the soil samples ranged from 60.0–140 mg kg$^{-1}$, with more than double the difference between the highest amount found in Metema soil and the lowest in Pawe soil. The highest and lowest concentrations of Cu found in the regional soil samples were reflected by the concentrations found in the corresponding rice samples.

The amount of Cu found in the soil samples is higher than the amount reported in soils from paddy fields in Korea (1.26–16.98 mg kg$^{-1}$) [28] and in China (1.57±0.69 mg kg$^{-1}$) [27]. The levels of Cu found in Fogera and Pawe soils were, however, lower than the permissible limits (100 mg kg$^{-1}$) set by FAO/WHO [21] for agricultural soils. Whereas, the amount of Cu determined in Metema soil was higher than the maximum permissible limit.

The measured concentration of Zn was in the range of 26.6–55.8 mg kg$^{-1}$, with the lowest found in soil from Pawe and the highest from Fogera rice farming areas. Soil samples from the rice farming areas of Fogera and Metema exhibited comparable concentrations of Zn to each other, while those from Pawe were significantly lower than the two areas. The determined concentrations of Zn in this study are lower than soils from rice cultivation fields in China (135 ±12.8 mg kg$^{-1}$) [27], garlic cultivation fields in Ethiopia (137–213 mg kg$^{-1}$) [32] and khat cultivation fields in Ethiopia (123–176 mg kg$^{-1}$) [10].

The average concentration of Ni determined in soil from Metema was 11.9 mg kg$^{-1}$ and from Fogera was 9.69 mg kg$^{-1}$, while it was below the LOQ of the method in soil from Pawe. Nickel was detected, but below the LOQ of the method, in all of the rice samples, while it was found in quantifiable amounts in the corresponding soils, except in Pawe. The amounts of Ni quantified in soils from Metema and Forera rice fields are lower than in soils from rice growing fields in China (32.14mg kg$^{-1}$) [36] and garlic cultivation fields in Ethiopia (87.5–124 mg kg$^{-1}$) [32].

In summary, except for Fe, the concentrations of all the determined elements varied significantly among the three regional rice-growing soils. Notably, soil samples from the rice cultivation fields of Pawe tend to contain lower amounts of metals than that from Fogera and Metema.

## Correlation analysis

Pearson correlation coefficients were calculated to assess the relationship between the total concentration of a metal present in the soil with that in the rice grain, as well as to assess whether a metal present in soil facilitates or interferes with the uptake of another metal.

Regarding the correlation of elements in the rice grains cultivated in the three regions, the two major essential metals, Ca and Mg, exhibited strong ($|r| > 0.9$) but negative correlation with each other. On the other hand, except for Cu, all of the determined trace essential metals were found to have a strong positive correlation with each other. Copper displayed no significant correlation with any of the trace essential metals. These elements all showed moderate ($|r|$ between 0.7 and 0.8) and negative correlation with Ca, while positively correlated with Mg.

Regarding the correlation of elements between the rice grains and soils from the three cultivation regions (Table 7), Cu exhibited a strong (r = 0.991) positive correlation between the rice grains and soils. This indicates that the more concentrated was Cu in the soil, the more its amount in the rice grain.

Generally, the order of the abundance of the elements in the soil is reflected by their abundance in the rice grains. One exception to this is Fe and Mg, where Fe exceeded in the soil while Mg was in rice. Mineral uptake by a plant is mainly a function of mineral concentration in soil, pH, the presence of ligands and competing metals and the botanical variety of the plant. As the same rice species (*O. sativa*) has been cultivated in the three regions, with all having comparable acidic soil pH conditions in the range of 4.90–4.95, the observed differences may be due to differences in the concentration of elements in soil, nature and amount of competing ligands and metals in soil.

The transfer characteristics of metal from soil to plant body can be assessed based on the transfer factor, which is the ratio of the concentration of the element in a plant part to that in

Table 7. Pearson correlation coefficients (r) between the concentrations of the elements determined in rice and soil samples.

| Element | Ca | Mg | Mn | Fe | Cu | Zn |
|---|---|---|---|---|---|---|
| r | 0.779 | -0.407 | -0.167 | 0.453 | 0.991 | 0.555 |

**Table 8. Soil-to-rice grain transfer factors for the elements quantified in rice and soil samples from the three major production regions of Ethiopia.**

| Element | Production region | | |
|---|---|---|---|
| | Fogera | Metema | Pawe |
| Ca | 0.10 | 0.09 | 0.17 |
| Mg | 0.79 | 0.59 | 0.88 |
| Mn | 0.14 | 0.06 | 0.18 |
| Fe | 0.01 | 0.01 | 0.01 |
| Cu | 0.40 | 0.30 | 0.20 |
| Zn | 0.50 | 0.46 | 0.92 |

the corresponding soil [37]. The transfer factor for the analyzed elements is calculated and presented in Table 8. All of the analyzed metals exhibited a transfer factor below 1, indicating that the rice plant only absorbs the metals from the soil but does not accumulate them in its seeds. Among the different analyzed metals, the highest transfer was observed for Zn and Mg with factors, respectively, of 0.92 and 0.88 at Pawe, followed by 0.50 and 0.79 at Fogera and 0.46 and 0.59 at Metema. Iron exhibited the lowest transfer factor of about 0.01 in all three regions.

## Conclusions

The levels of metals (Ca, Mg, Fe, Mn, Zn, Cu, Ni, Cr, Cd and Pb) in rice (*Oryza sativa*) grains, and the corresponding growing soil, cultivated in Metema, Fogera and Pawe, Ethiopia, were determined. Magnesium was the most abundant of the elements in rice, while it was Fe in soil. The elements Cr, Cd, Pb and Ni were below the limits of quantifiable of the method employed, except Ni in soils from Metema and Fogera rice fields. The concentrations of the elements, except Zn in rice and Fe in soil, varied significantly among the different growing regions. Generally, except for the reversal of Fe and Mn, the order of the abundance of the elements in the soil was reflected by their abundance in the rice grains. However, elemental concentrations were higher in soil than in rice grains, with rice-to-soil ratios of less than one, indicating the absence of bioaccumulation by rice grains. Additionally, only Cu exhibited a strong positive correlation between rice grains and soils across the three growing regions.

## Acknowledgments

The authors thank University of Gondar for giving access to its laboratory facility.

## Author Contributions

**Conceptualization:** Bewketu Mehari.

**Data curation:** Abebe Desalew.

**Investigation:** Abebe Desalew.

**Methodology:** Bewketu Mehari.

**Supervision:** Bewketu Mehari.

**Writing – original draft:** Abebe Desalew.

**Writing – review & editing:** Bewketu Mehari.

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
