## [Decision Letter · Decision Letter 0]

12 May 2023

PONE-D-23-10177Variations in Elemental Composition of Rice with Different Cultivation Areas of EthiopiaPLOS ONE

Dear Dr. Mehari,

Thank you for submitting your manuscript to PLOS ONE. After careful consideration, we feel that it has merit but does not fully meet PLOS ONE’s publication criteria as it currently stands. Therefore, we invite you to submit a revised version of the manuscript that addresses the points raised during the review process.

We look forward to receiving your revised manuscript.

Kind regards,

Andrea Mastinu

Academic Editor

PLOS ONE

Journal Requirements:

3. We note that Figure (1) in your submission contain copyrighted images. All PLOS content is published under the Creative Commons Attribution License (CC BY 4.0), which means that the manuscript, images, and Supporting Information files will be freely available online, and any third party is permitted to access, download, copy, distribute, and use these materials in any way, even commercially, with proper attribution. For more information, see our copyright guidelines: http://journals.plos.org/plosone/s/licenses-and-copyright.

1. You may seek permission from the original copyright holder of Figure (1) to publish the content specifically under the CC BY 4.0 license. 

Reviewers' comments:

Reviewer's Responses to Questions

**Comments to the Author**

1. Is the manuscript technically sound, and do the data support the conclusions?

Reviewer #1: Yes

Reviewer #2: Yes

2. Has the statistical analysis been performed appropriately and rigorously? 

Reviewer #1: Yes

Reviewer #2: Yes

3. Have the authors made all data underlying the findings in their manuscript fully available?

Reviewer #1: Yes

Reviewer #2: Yes

4. Is the manuscript presented in an intelligible fashion and written in standard English?

Reviewer #1: Yes

Reviewer #2: Yes

5. Review Comments to the Author

Reviewer #1: The literature survey is not comprehensive. The following literature should be reviewed and cite in the Introduction:

Physical properties and chemical composition of three Ethiopian rice (Oryza sativa Linn.) varieties compared to tef [Eragrostis tef (Zucc.) Trotter] grain. https://doi.org/10.51745/najfnr.3.6.180-185

Effect of rice variety and location on nutritional composition, physicochemical, cooking and functional properties of newly released upland rice varieties in Ethiopia. Cogent Food And Agriculture 2021, 7(1):1945281. DOI: 10.1080/23311932.2021.1945281

Physicochemical, Nutritional Composition, Cooking, and Functional Properties of Newly Introduced Low Land Rice Varieties Grown in Ethiopia. Philippine Journal of Science 2020, 150(3):923-934. DOI: 10.56899/150.03.27

Global Geographical Variation in Elemental and Arsenic Species Concentration in Paddy Rice Grain Identifies a Close Association of Essential Elements Copper, Selenium and Molybdenum with Cadmium. Exposure and Health (2022). https://doi.org/10.1007/s12403-022-00504-1

Table 1 should be deleted. The wavelength column should be incorporated in Table 5.

Comparison of levels of metals in this study with other studies reported from Ethiopia and other African countries should be given.

References require carefully checking and editing.

Ref. [9], the article title should be corrected as in the other references.

Ref. [11], the article title should be corrected as in the other references.

Ref. [16], the article title should be corrected as in the other references.

Ref. [19], the article title should be corrected as in the other references.

Ref. [23], the article title should be corrected as in the other references.

Ref. [32], Bulletene should be Bulletin

Reviewer #2: Dear Authors

your work is very interesting but still need some improvement to be suitable for publication. Please find attached file for comments and modifications.

The plant Latin name must added in the name of manuscript in full details (add author name) Oryza sativa L.

Plant name along the manuscript must be in a short form as O. sativa.

Some editing errors must be corrected.

Please determine the area of the studied areas.

The family name of rice must be updated to Poaceae instead of Gramineae.

6. PLOS authors have the option to publish the peer review history of their article (what does this mean?). If published, this will include your full peer review and any attached files.

Reviewer #1: No

Reviewer #2: No

---

## [Author Response · Author response to Decision Letter 0]

12 Jul 2023

PONE-D-23-10177

Variations in Elemental Composition of Rice with Different Cultivation Areas of Ethiopia

Andrea Mastinu

Journal Requirements:

Author Response: We thank the editor for handling our manuscript favorably. We have followed PLOS ONE’s style requirements.

Author Response: Samples were collected from private farmlands, after permission was granted by the farmers to access their fields. This information is now provided in the method section.

3. We note that Figure (1) in your submission contain copyrighted images. All PLOS content is published under the Creative Commons Attribution License (CC BY 4.0), which means that the manuscript, images, and Supporting Information files will be freely available online, and any third party is permitted to access, download, copy, distribute, and use these materials in any way, even commercially, with proper attribution. For more information, see our copyright guidelines: http://journals.plos.org/plosone/s/licenses-and-copyright.

Author Response: The map is constructed from a freely available world shape file. This is now indicated in the figure caption.

(https://datacatalog.worldbank.org/search/dataset/0038272/World-Bank-Official-Boundaries)

1. You may seek permission from the original copyright holder of Figure (1) to publish the content specifically under the CC BY 4.0 license. 

Review Comments to the Author

Reviewer #1: 

Author Response: We thank Reviewer #1 for reviewing our manuscript favorably.

The literature survey is not comprehensive. The following literature should be reviewed and cite in the Introduction:

Physical properties and chemical composition of three Ethiopian rice (Oryza sativa Linn.) varieties compared to tef [Eragrostis tef (Zucc.) Trotter] grain. https://doi.org/10.51745/najfnr.3.6.180-185

Effect of rice variety and location on nutritional composition, physicochemical, cooking and functional properties of newly released upland rice varieties in Ethiopia. Cogent Food And Agriculture 2021, 7(1):1945281. DOI: 10.1080/23311932.2021.1945281

Physicochemical, Nutritional Composition, Cooking, and Functional Properties of Newly Introduced Low Land Rice Varieties Grown in Ethiopia. Philippine Journal of Science 2020, 150(3):923-934. DOI: 10.56899/150.03.27

Global Geographical Variation in Elemental and Arsenic Species Concentration in Paddy Rice Grain Identifies a Close Association of Essential Elements Copper, Selenium and Molybdenum with Cadmium. Exposure and Health (2022). https://doi.org/10.1007/s12403-022-00504-1

Author Response: We have reviewed the suggested articles and the relevant information have been incorporated in the introduction section.

Table 1 should be deleted. The wavelength column should be incorporated in Table 5.

Author Response: Table 1 is deleted and wavelength column is incorporated in Table 5.

Comparison of levels of metals in this study with other studies reported from Ethiopia and other African countries should be given.

Author Response: We have compared the results of this study with other reported values from Ethiopia and some other countries in the results and discussion section.

References require carefully checking and editing.

Ref. [9], the article title should be corrected as in the other references.

Ref. [11], the article title should be corrected as in the other references.

Ref. [16], the article title should be corrected as in the other references.

Ref. [19], the article title should be corrected as in the other references.

Ref. [23], the article title should be corrected as in the other references.

Ref. [32], Bulletene should be Bulletin

Author Response: All the references have been carefully checked and edited.

Reviewer #2: Dear Authors

Your work is very interesting but still need some improvement to be suitable for publication.

Author Response: We thank Reviewer #2 for reviewing our manuscript favorably.

Please find attached file for comments and modifications.

Author Response: All the comments and modifications provided directly in the attached manuscript are incorporated.

The plant Latin name must added in the name of manuscript in full details (add author name) Oryza sativa L.

Author Response: The plant Latin name has been revised as Oryza sativa L.

Plant name along the manuscript must be in a short form as O. sativa.

Author Response: We have used the short form as O. sativa, after introducing the full name, along the manuscript.

Some editing errors must be corrected: 

Author Response: We have thoroughly revised the manuscript according to the comments and suggestions provided in the attached manuscript. 

Please determine the area of the studied areas.

Author Response: We have provided the names of the areas from which the samples were collected.

The family name of rice must be updated to Poaceae instead of Gramineae.

Author Response: We have updated the family name as Poaceae.

---

## [Decision Letter · Decision Letter 1]

25 Jul 2023

PONE-D-23-10177R1Variations in Elemental Composition of Rice (Oryzasativa L.) with Different Cultivation Areas of EthiopiaPLOS ONE

Dear Dr. Mehari,

Thank you for submitting your manuscript to PLOS ONE. After careful consideration, we feel that it has merit but does not fully meet PLOS ONE’s publication criteria as it currently stands. Therefore, we invite you to submit a revised version of the manuscript that addresses the points raised during the review process.

We look forward to receiving your revised manuscript.

Kind regards,

Andrea Mastinu

Academic Editor

PLOS ONE

Journal Requirements:

Reviewers' comments:

Reviewer's Responses to Questions

**Comments to the Author**

1. If the authors have adequately addressed your comments raised in a previous round of review and you feel that this manuscript is now acceptable for publication, you may indicate that here to bypass the “Comments to the Author” section, enter your conflict of interest statement in the “Confidential to Editor” section, and submit your "Accept" recommendation.

Reviewer #1: All comments have been addressed

Reviewer #2: (No Response)

2. Is the manuscript technically sound, and do the data support the conclusions?

Reviewer #1: Yes

Reviewer #2: Yes

3. Has the statistical analysis been performed appropriately and rigorously? 

Reviewer #1: Yes

Reviewer #2: Yes

4. Have the authors made all data underlying the findings in their manuscript fully available?

Reviewer #1: Yes

Reviewer #2: Yes

5. Is the manuscript presented in an intelligible fashion and written in standard English?

Reviewer #1: Yes

Reviewer #2: Yes

6. Review Comments to the Author

Reviewer #1: The manuscript has been revised appropriately according to my comments. The revised manuscript is acceptable for publication.

Reviewer #2: The manuscript be better just few editing corrections are needed. Please follow attached file for corrections

7. PLOS authors have the option to publish the peer review history of their article (what does this mean?). If published, this will include your full peer review and any attached files.

Reviewer #1: **Yes: **Bhagwan Singh Chandravanshi

Reviewer #2: No

---

## [Author Response · Author response to Decision Letter 1]

29 Jul 2023

Journal Requirements:

Author Response: We thank the editor for handling our manuscript favorably. We have reviewed the reference list to ensure that it is complete and correct.

Reviewers' comments:

Reviewer #1: The manuscript has been revised appropriately according to my comments. The revised manuscript is acceptable for publication.

Author Response: We thank Reviewer #1 for reviewing our manuscript favorably.

Reviewer #2: The manuscript be better just few editing corrections are needed. Please follow attached file for corrections

Author Response: We thank Reviewer #2 for reviewing our manuscript favorably. We have incorporated all the comments and modifications provided directly in the attached manuscript.

---

## [Editor Report · Decision Letter 2]

2 Aug 2023

Variations in Elemental Composition of Rice (Oryzasativa L.) with Different Cultivation Areas of Ethiopia

PONE-D-23-10177R2

Dear Dr. Mehari,

We’re pleased to inform you that your manuscript has been judged scientifically suitable for publication and will be formally accepted for publication once it meets all outstanding technical requirements.

Kind regards,

Andrea Mastinu

Academic Editor

PLOS ONE
---

## [Editor Report · Acceptance letter]

11 Oct 2023

PONE-D-23-10177R2 

Variations in Elemental Composition of Rice (*Oryza sativa L.*) with Different Cultivation Areas of Ethiopia 

Dear Dr. Mehari:

I'm pleased to inform you that your manuscript has been deemed suitable for publication in PLOS ONE. Congratulations! Your manuscript is now with our production department. 

Kind regards, 

on behalf of

Dr. Andrea Mastinu 

Academic Editor

PLOS ONE